# The vaginal Torquetenovirus titer varies with vaginal microbiota composition in pregnant women

Tania Regina Tozetto-Mendoza[1]*, Maria C. Mendes-Correa[1], Antonio F. Moron[2], Larry J. Forney[3], Iara M. Linhares[4], Almir Ribeiro da Silva, Jr.[1], Layla Honorato[1], Steven S. Witkin[1,5]

1 Faculdade de Medicina da Universidade de São Paulo, Departament of Infectious Diseases, Laboratório de Investigação Médica em Virologia (LIM52), Instituto de Medicina Tropical de São Paulo, São Paulo, Brazil, 2 Department of Obstetrics, Federal University of São Paulo, São Paulo, Brazil, 3 Department of Biological Sciences and Institute for Bioinformatics and Evolutionary Studies, University of Idaho, Moscow, Idaho, United States of America, 4 Department of Gynecology and Obstetrics, University of São Paulo Medical School, São Paulo, Brazil, 5 Department of Obstetrics and Gynecology, Weill Cornel Medicine, New York, New York, United States of America

* tozetto@usp.br

**Data Availability Statement:** All relevant data are within the paper and supporting information.

## Abstract

Torquetenovirus (TTV) is a nonpathogenic endogenous virus whose abundance varies with the extent of immune system activation. We determined if the TTV titer in the vagina of pregnant women was associated with vaginal microbiota composition and levels of compounds in vaginal secretions. Vaginal TTV and microbiota composition in 494 second trimester pregnant women were identified by gene amplification and analysis. Vaginal matrix metalloproteinases (MMPs), tissue inhibitors of MMP (TIMP) and lactic acid isomers were measured by ELISA. Dominance was defined as the relative abundance of a specific bacterium or species at >50% of the total number of bacteria identified. Clinical data were obtained by chart review. The median $\log_{10}$ TTV titer was lowest when *Lactobacillus* species other than *L. iners* were dominant (<1.0) as compared to when *L. iners* (4.1, p = 0.0001), bacteria other than lactobacilli (4.5, p = 0.0016) or no bacterium (4.7, p = 0.0009) dominated. The TTV titer was inversely proportional to *L. crispatus* abundance (p<0.0001) and directly proportional to levels of *G. vaginalis* (p = 0.0008) and *L. iners* (p = 0.0010). The TTV titer was proportional to TIMP-1, TIMP-2, MMP-8 and MMP-9 abundance (p≤0.0002) and inversely proportional to the level of D-lactic acid (p = 0.0024). We conclude that the association between variations in the TTV titer and the relative abundance of specific bacterial species and vaginal compounds indicates that local changes in immune status likely influence vaginal fluid composition.

## Introduction

Torquetenovirus (TTV), a small single-stranded DNA virus and a member of the *Alphatorquetenovirus* genus and the Anelloviridae family [1], is present in the circulation and body fluids of healthy individuals [1,2]. Its ubiquitous presence and lack of association with pathogenic conditions leads to its characterization as a non-pathogenic endogenous virus [1]. However,

**Funding:** The analysis of vaginal samples for TTV was supported by departmental funds in the Laboratório de Investigaçao Médica em Virologia (LIM-52), Faculdade de Medicina, Universidade de São Paulo. The collection of study samples, microbiome testing and collection of demographic data and pregnancy outcome in this cohort was supported by grants from the Bill and Melinda Gates Foundation, Brazilian Ministry of Health (DECIT) and the Brazilian National Research Council (CNPq – grant 401626/2013-0). Sequence data collection and analyses were performed by the IBEST Genomics Resources Core at the University of Idaho, which is supported in part by NIH COBRE grant P30GM103324. The funders had no role in study design, data collection and analysis, decision to publish, or preparation of the manuscript.

**Competing interests:** The authors have declared that no competing interests exist.

the circulating TTV concentration becomes markedly elevated in individuals whose immune system is compromised, due either to the intake of immunosuppressive medications following organ transplantation [3,4] or as a consequence of HIV infection [5]. This has led to the suggestion that measurement of the TTV titer may be a sensitive indicator of immune status [6].

In pregnant women, TTV has been identified in peripheral blood [7] and in cervical [8] and vaginal [9,10] secretions. In the above studies, the presence of TTV was not associated with any pregnancy-related pathology.

One of four *Lactobacillus* species, *L. crispatus*, *L. iners*, *L. gasseri* and *L. jensenii*, are usually the most abundant species in the vaginas of pregnant women worldwide [11]. In a minority of women other species of lactobacilli, *Gardnerella vaginalis* or other anaerobic bacteria may predominate, while in others no single bacterial species is present at >50% of the total population [12–15]. Differences in bacterial species abundance in the vagina during pregnancy have been shown to be associated with alterations in the concentration of compounds in the vagina [15]. For example, the D- isomer of lactic acid is produced by all lactobacilli species present in the vagina except *L. iners;* non-lactobacillus species in the vagina also do not produce D-lactic acid. L-lactic acid is produced by all vaginal lactobacilli, including *L. iners* [15,16]. Conversely, matrix metalloproteinases (MMPs) and tissue inhibitors of metalloproteinases (TIMPs) are elevated when *L. iners* or non-lactobacilli predominate in the vagina. Levels of these compounds are markedly reduced when vaginal D-lactic acid concentrations are high and when *L. crispatus* is dominant [15,17–19].

The aim of the present study was to evaluate, in a large population of pregnant women in Brazil, whether variations in the TTV titer in vaginal secretions were associated with changes in the vaginal microbiota and concentrations of compounds in vaginal secretions. Demonstration of these associations would indicate that the TTV level reflects alterations in the vaginal environment in pregnant women.

## Methods

### Study population

The study was retrospective and included women who were both at high or low risk for an adverse pregnancy outcome based on past history and current cervical length determination. Inclusion was based only on the availability of vaginal samples for analysis and not on any outcome or historical variables. We enrolled 494 pregnant women who were seen for a routine second trimester evaluation at three hospital-based outpatient clinics in Brazil: The Federal University of São Paulo, The University of Jundiai Medical School, and The Federal University of Ceara. Exclusion criteria were signs or symptoms consistent with a genital tract infection, antibiotic usage in the previous two weeks or history of an endocrine or immunological disorder. The women in our study who were at elevated risk for preterm birth, due to a history of this occurrence or whose cervix was ≤25mm, were treated with progesterone to reduce their risk, as previously reported [15]. Thus, we were unable to analyze a possible association between preterm birth and TTV titer. In all cases the progesterone was administered after the vaginal samples were collected for our analysis.

The study was approved by the Institutional Review Board of the Federal University of São Paulo, approval number 1.095.610 June 17, 2015. All subjects provided written informed consent.

### Sample collection

The methods used to collect vaginal samples from these subjects were described previously [15]. Briefly, samples were obtained from the posterior vagina with a cotton swab that was

placed in a sterile tube containing 1 ml of sterile phosphate-buffered saline and vigorously shaken. The tube was centrifuged, and aliquots of the supernatant were stored at –80˚C before analysis. A second set of samples was collected (Copan ESwab sample collection system, Fisher Scientific, Pittsburgh, PA), for microbiome analysis.

### TTV titer analysis

Viral DNA was extracted from thawed vaginal supernatants and purified using the MVXA-P096 FAST kit and automatized extractor (EXTRACT 96, Loccus, Cotia, São Paulo, Brazil), according to the manufacturer's instructions. Suitability of the DNA for viral DNA amplification was evaluated by first testing the samples for amplification of the gene coding for beta-globin. This was successful in every case and indicated that the DNA was intact and that our preparations did not contain PCR inhibitors. In addition, specifically concerning the stability of the TTV DNA, we obtained a suitable amplification profile by real time PCR in every TTV positive samples and controls (S1 Fig). This indicates that the viral DNA was quantitatively stable.

The primers and probe sequences for DNA TTV amplification were: Forward primer 5′-GTGCCGIAGGTGAGTTTA-3′; Reverse primer 5′-AGCCCGGCCAGTCC-3′; Probe: FAM5′-TCAAGGGGCAATTCGGGCT-3' [9,20]. For the real time quantitative PCR reaction (qPCR), a standard curve was generated, as described previously, with known amounts of the synthetic oligonucleotide: 5'TTCGTAGCCCGGCCAGTCC CGTAT AGCCCGAATTGCCCCTTGA ATGCGT TAAACTCACCTTCGGCAC CTGATA -3' [10]. The qPCR mix was prepared with forward and reverse primers at 250 nM and the probe at 62 nM and a standard input of $\sim 100$ ng of DNA template per reaction in 12.5 μl of 2X TaqMan™ Universal Master Mix (Thermo Fisher Scientific, Warrington, UK). Thermocycling condition consisted of two initial heat activation steps of 50˚C for 2 minutes and 95˚C for 15 seconds, followed by 50 cycles of 15 seconds at 95˚C and 1 minute at 60˚C, in a Quantstudio™ 5 instrument. The data were analyzed using QuantStudio Design & Analysis Software v.1.4.1. The limit of sensibility of TTV qPCR was 40 copies/ml at the > 95% detection rate [10].

### Vaginal compound analysis

Vaginal levels of the D- and L-lactic acid isomers were quantitated by colorimetric assays using the EnzyChrom D-lactic acid and L-lactic acid kits (BioAssay Systems, Haywood, CA). The levels of tissue inhibitor of matrix metalloproteinase (TIMP)-1 and TIMP -2, matrix metalloproteinase (MMP)-2, MMP-8, and MMP-9 in the vaginal fluid supernatants were determined using commercial ELISA kits (R&D Systems, Minneapolis, MN). Values were determined by reference to a standard curve that was included with each set of samples.

### Vaginal microbiota analysis

The protocol used to determine vaginal microbiota composition by amplification and analysis of the V1-V3 region of the gene coding bacterial 16S ribosomal RNA, along with the outcomes in the majority of these women, have been reported previously [15,21]. Briefly, bacterial genomic DNA was isolated with a QIAamp DNA minikit and the V1-V3 regions of bacterial 16S ribosomal RNA genes were amplified using specific primers flanking the variable regions. The production of specific amplicons were generated by two consecutive rounds of amplification that also included the attachment of sample barcodes and sequencing adapters. DADA2 software (v1.8) was used to identify distinct sequence variants (DSVs) and to remove chimeras. All DSVs were classified to the genus level using the RDP naive Bayesian classifier (v11.5) in

combination with SILVA reference database and assigned to the appropriate species using SPINGO software.

### Statistics

Differences in continuous variables were analyzed by the non-parametric Kruskal-Wallis and Mann-Whitney tests, as appropriate. Categorical variables were analyzed by Fisher's exact test. Associations between the percent abundance of a specific bacterium in the vaginal microbiota of individual women or the concentration of compounds in vagina secretions and the TTV titer were analyzed by the Spearman rank correlation test. In all analyses we employed the Bonferroni correction for multiple testing and, thus, a p value <0.005 was considered significant. All analyses utilized GraphPad Prism 9 software (San Diego, CA).

## Results

Characteristics of the study population are shown in Table 1. Mean gestational age at sample collection and analysis was 21.0 weeks. The mean maternal age was 30.4 years, median gravity and parity were 2.0 and 1.0, respectively and mean gestational age at delivery was 38.2 weeks. Most of the women were either White (49.4%) or mixed race (41.7%).

The association between bacterial abundance in the vaginal microbiota, defined as constituting >50% of the total bacteria identified, and the median $\log_{10}$ TTV titer in vaginal secretions is presented in Table 2. The TTV titer when each individual bacterium was dominant is presented in supplementary S1 Table. When the *Lactobacilli L. crispatus*, *L. jensenii*, *L. gasseri*, *L. delbreuckii* and *L. vaginalis* were most abundant the median $\log_{10}$ TTV titer was <1.0. This was significantly lower than when *L.* iners ($\log_{10}$ = 4.1), or non-lactobacilli (*G. vaginalis*, *Lachnospiraceae*, *Shuttleworthia*, $\log_{10}$ = 4.5) were most abundant or when no bacterium was present at >50% of the total ($\log_{10}$ = 4.7), (p = 0.0001, Kruskal-Wallis test). As the proportion of *L. crispatus* as a component of the microbiome decreased the TTV titer increased (Spearman rho = - 0.1818, p <0.0001) In marked contrast, the TTV titer increased as the proportions of *G. vaginalis* (Spearman rho = 0.1557, p = 0.0008) or *L. iners* (Spearman rho = 0.1129, p = 0.0010) increased. TTV was below the level of detection in 50%, 28.9%, 31.3%, and 16.9% of the samples when lactobacilli species except for *L. iners*, *L. iners*, non-lactobacilli or no bacteria was dominant, respectively. Differences in the percent non-detectable TTV when lactobacilli minus *L. iners* was most abundant and when each the other three categories were most abundant was significant (p≤0.0097, Fisher's exact test).

**Table 1. Characteristics of study population of 494 women.**

| Characteristic | |
|---|---|
| Mean age (SD) | 30.4 (7.0) years |
| Median gravidity (range) | 2.0 (1–10) |
| Median parity (range) | 1.0 (0–7) |
| Mean body mass index (SD) | 27.5 (5.8) kg/m$^2$ |
| Mean gestational age sample (SD) | 21.0 (1.2) weeks |
| Mean gestational age delivery (SD) | 38.2 (2.6) weeks |
| Mean baby birthweight (SD) | 3132 (569) grams |
| White | 49.4% |
| Mixed race | 41.7% |
| Black | 8.9% |

SD (Standard deviation).

**Table 2. Association between the TTV titer in vaginal secretions and the dominant bacteria in the vaginal microbiome.**

| Dominant bacteria[a] | No. tested | Median $\log_{10}$ TTV titer (interquartile range)[b] |
|---|---|---|
| *Lactobacilli*[c] | 236 | <1.0 (<1.0, 4.5) |
| *L. iners* | 158 | 4.1 (<1.0, 5.1) |
| Non-*lactobacilli*[d] | 64 | 4.5 (<1.0, 5.7) |
| None[e] | 34 | 4.7 (<1.0, 5.0) |

[a]relative abundance >50% of the total number of bacteria identified.

[b]difference between the four categories tested p < 0.0001, Kruskal-Wallis test.

[c]*L. crispatus* N = 207, *L. jensenii* N = 15, *L. gasseri* N = 12, *L. delbreuckii* N = 1, *L. vaginalis* N = 1.

[d]*G. vaginalis* N = 59, *Shuttleworthia* N = 4, *Lachnospiraceae* N = 1.

[e]No bacterium was present at >50% of the total number of bacteria detected.

MMP-2, MMP-8, MMP-9, TIMP-1, TIMP-2 and D- and L-lactic acid were identified in every vaginal sample. Their concentrations and associations between the vaginal median $\log_{10}$ TTV titer and the concentration of each of these compounds in vaginal secretions are shown in Table 3. The higher the TTV titer the higher was the concentration of TIMP-1 (p = 0.0002), TIMP-2 (p<0.0001), MMP-8 (p < 0.0001) and MMP-9 (p < 0.0001), but not MMP-2 (p = 0.2854) in the vagina. In contrast, the TTV titer was inversely related to the vaginal concentration of the D-lactic acid isomer (p = 0.0024), but not to L-lactic acid (p = 0.0943, Spearman rank correlation test).

We were unable to accurately determine a possible association between TTV titer and preterm birth since women at risk for this outcome, due to the presence of a cervical length <25mm or a history of preterm birth, were treated with progesterone, after vaginal sample collection. However, using treatment with progesterone as a surrogate marker for elevated risk of preterm birth, there was no difference in the median (interquartile range) $\log_{10}$ TTV titer between the 62 women who received progesterone, 4.1 (<1,0,5.0) and the 432 women who were untreated, 3.7 (<1.0,5.0) (p = 0.2323, Mann-Whitney test). There were no associations between the median $\log_{10}$ TTV titer and race (p = 9183, Kruskal-Wallis test) (S2 Table) or maternal age (Spearman rho = - 0.0443, p = 0.3268). The TTV titers from women at each of the three study sites were also equivalent (p = 0.3268, Mann-Whitney test) (S3 Table).

**Table 3. Compounds in vaginal secretions and their association with the $\log_{10}$ TTV titer.**

| Compound (unit) | Median (ng/ml) (interquartile range) | Association with TTV titer (Spearman rho)[a] | p value |
|---|---|---|---|
| TIMP-1 | 1.5 (0.3, 5.0) | 0.1751 | **0.0002** |
| TIMP-2 | 4.6 (2.4 8.7) | 0.1795 | < **0.0001** |
| MMP-2 | 0.8 (<0.2–3.8) | 0.0485 | 0.2854 |
| MMP-8 | 33.8 (13.0, 52.6) | 0.1844 | < **0.0001** |
| MMP-9 | 19.4 (9.9, 27.1) | 0.3071 | < **0.0001** |
| D-lactic acid | 0.75 (0.10, 2.29)[b] | − 0.1380 | **0.0024** |
| L-lactic acid | 1.09 (0.42, 2.38)[b] | 0.0756 | 0.0943 |

TIMP (tissue inhibitor of matrix metalloproteinase).

MMP (matrix metalloproteinase).

[a]Analysis was by the Spearman rank correlation test.

[b]Values are mM.

## Discussion

TTV was detectable in the vaginas in the majority of second trimester pregnant women from three sites in Brazil. The median $\log_{10}$ TTV titer was lowest when lactobacilli species other than *L. iners* were present at the highest relative abundance. The median titer was elevated $> 4$ logs when *L. iners* or bacteria other than lactobacilli species were most abundant, or when no bacterial species was present at $>50\%$ of the total. The TTV titer was inversely associated with the relative abundance of *L. crispatus* in the vaginal microbiome and positively associated with the levels of *L. iners* and *G. vaginalis*. The vaginal D-lactic acid level was also inversely related to the TTV titer, while TIMP-1, TIMP-2, MMP-8 and MMP-9 levels all increased in proportion to the TTV titer. These findings strongly suggest that the vaginal TTV titer varies with local conditions in the vagina. Thus, the vaginal TTV titer appears to be an additional indicator of the local vaginal status in pregnant women.

The mechanism associated with variations in the vaginal TTV titer in individual women remain to be determined. Since TTV replication is believed to preferentially occur in activated lymphoid cells [22–24], and immune system activation in the vagina is at its lowest level when *L. crispatus* dominates at that site [17], it is reasonable to speculate that the low TTV titer is a reflection of the scarcity of lymphoid cells or their cytokine products when *L. crispatus* or other *Lactobacilli* species except *L. iners* are predominate.

Strengths of the present investigation are the large number of subjects analyzed and the consistency of the findings from samples obtained from women at three different sites. The association between TTV titer and vaginal microbiota composition are also consistent with results obtained from a much more limited study of women in the United States [10]. Study limitations must be acknowledged. Maternal blood was not available from our study subjects and so we were not able to compare TTV concentrations in the vagina to those in blood or comment whether or not the vaginal TTV originated in the systemic circulation. It is still of interest to point out that the range of TTV titers we identified in the vagina parallels the levels reported to be present in peripheral blood (2–8 $\log_{10}$ copies/ml) [22]. In addition, only a limited number of compounds present in the vagina were tested. We are, thus, unable to comment on the range of compounds that may be associated, either positively or negatively, with the TTV titer. In the absence of an evaluation of lymphocyte or cytokine levels in the vagina our hypothesis that the observed TTV variations are due to differences in local immunity remains speculative. The low abundance in our sample of bacteria such as *L. jensenii*, *L. gasseri*, *Leptospiraceae* and others makes it difficult to accurately assess their association with the TTV titer. Lastly, as mentioned in Methods, progesterone treatment of women who were at risk for preterm birth obviated our ability to assess the relationship between TTV titer and gestational age at delivery. As shown in Table 1, the overwhelming percentage of women in our study delivered at term ($\geq 37$ weeks gestation). However, our finding that the TTV titer in midtrimester pregnant women who were treated with progesterone because they were deemed to be at elevated risk for preterm birth was not different from pregnant women who were not at high risk for an adverse outcome is consistent with prior studies [7–10] that did not find a relationship between TTV titer and adverse pregnancy outcome. We wish to stress that all vaginal samples in the present study were obtained prior to progesterone treatment, and so its utilization in some women had no influence on our results.

In conclusion, the TTV titer in vaginal secretions during the mid-trimester of pregnancy varies with the relative abundance of different bacteria in the vaginal microbiota as well as with the concentration of select compounds that are present at that site. Explanations for differences in the patterns of vaginal colonization in individual women have been advanced [17–21], but the aetiology still remain poorly understood. The novel observation of an association

between the vaginal TTV titer and differential bacterial dominance in the vagina strongly suggests that local changes in immune status influence vaginal fluid composition. Experimentation to identify the specific immune variations may help in defining the local conditions that favor the proliferation of individual species of bacteria. This may lead to development of protocols to alter the vaginal environment and improve women's health.

## Supporting information

**S1 Fig. Amplification plot–DNA TTV quantitative real-time polymerase chain reaction (qPCR) in vaginal secretions.** The TTV titer was obtained by using a qPCR based on the Taq-Man™ Universal PCR master mix protocol (Thermo Fisher Scientific, Warrington, UK), a purified DNA template (~100 ng/reaction) in a final volume of 25µl. The condition for thermocycling was 50oC for 2 minutes, 95oC for 15 seconds and followed by 50 cycles of 95oC for 15 seconds and 60oC for 1 minute, in a Quantstudio™ 5 thermocycler. The data were analyzed using QuantStudio Design & Analysis Software v.1.4.1. The analytical sensibility (LOD>95%) of TTV qPCR is 40 copies/ml. Cycle number is shown on the x-axis and change in florescent intensity (ΔRn) is shown on the y-axis; the horizontal line indicates the threshold setting. NC, Non-template control or Negative control; CP, TTV positive genital fluid samples.
(TIF)

**S1 Table. Association between the TTV titer in vaginal secretions and the dominant bacterium in the vaginal microbiome.**
(PDF)

**S2 Table. Association between TTV titer in vaginal secretions and race.**
(PDF)

**S3 Table. TTV titer by site.**
(PDF)

## Acknowledgments

The authors thank Lucy Santos Vilas Boas, Silvia Helena Lima, Anderson Vicente de Paula, Wilton Santos Freire and Marli Estevam de Paula for excellent technical support.

## Author Contributions

**Conceptualization:** Tania Regina Tozetto-Mendoza, Steven S. Witkin.

**Data curation:** Antonio F. Moron, Larry J. Forney, Almir Ribeiro da Silva, Jr., Steven S. Witkin.

**Formal analysis:** Tania Regina Tozetto-Mendoza, Maria C. Mendes-Correa, Antonio F. Moron, Larry J. Forney, Iara M. Linhares, Steven S. Witkin.

**Funding acquisition:** Maria C. Mendes-Correa.

**Investigation:** Maria C. Mendes-Correa, Iara M. Linhares, Almir Ribeiro da Silva, Jr., Layla Honorato, Steven S. Witkin.

**Methodology:** Tania Regina Tozetto-Mendoza, Maria C. Mendes-Correa, Antonio F. Moron, Larry J. Forney, Iara M. Linhares, Almir Ribeiro da Silva, Jr., Layla Honorato, Steven S. Witkin.

**Supervision:** Maria C. Mendes-Correa, Antonio F. Moron, Larry J. Forney, Iara M. Linhares, Steven S. Witkin.

**Validation:** Steven S. Witkin.

**Visualization:** Steven S. Witkin.

**Writing – original draft:** Tania Regina Tozetto-Mendoza, Maria C. Mendes-Correa, Antonio F. Moron, Larry J. Forney, Iara M. Linhares, Almir Ribeiro da Silva, Jr., Layla Honorato, Steven S. Witkin.

**Writing – review & editing:** Tania Regina Tozetto-Mendoza, Maria C. Mendes-Correa, Antonio F. Moron, Larry J. Forney, Iara M. Linhares, Almir Ribeiro da Silva, Jr., Layla Honorato, Steven S. Witkin.

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
