## [Decision Letter · Decision Letter 0]

2 Nov 2021

PONE-D-21-30550The vaginal Torquetenovirus titer varies with vaginal microbiome composition in pregnant womenPLOS ONE

Dear Dr. Tozetto-Mendoza,

Thank you for submitting your manuscript to PLOS ONE. After careful consideration, we feel that it has merit but does not fully meet PLOS ONE’s publication criteria as it currently stands. Therefore, we invite you to submit a revised version of the manuscript that addresses the points raised during the review process. I strongly agree with the reviewers' comments that the conclusions are overstated, and several speculative comments are made which are not supported by the data. In a field like this where data are few, I would encourage you to make more factual statements about the findings about correlation between TTV titer and microbial communities, and not speculate about the causal relationship between them - especially based on cross-sectional data.  It might be worth a little more discussion of TTV and its link to immune competence, and how you think TTV could be used as a biomarker IF your results are confirmed. 

We look forward to receiving your revised manuscript.

Kind regards,

Caroline Mitchell

Academic Editor

PLOS ONE

Additional Editor Comments (if provided):

I have a few additional specific comments to add to the reviewers:

Line 64: Goal is to determine “…if the TTV titer in vaginal secretions was predictive of vaginal microbiome composition and pregnancy-related parameters.”

Line 90: “All samples were suitable for DNA amplification by using internal control.” – it is unclear exactly what this means? Was a region amplified to show there was no inhibition? Was the quality and quantity of nucleic acid measured?

Line 180-185: “We propose that a low or undetectable vaginal TTV titer is a biomarker for lower genital tract quiescence during the second trimester of pregnancy, similar to when L. crispatus is dominant at that site.” – How can you separate the quiescence and L. crispatus dominance? It doesnt' seem that you have shown that the two are separately associated with TTV

Line 191-193: “Thus, a low vaginal TTV titer appears to parallel L. crispatus vaginal dominance as an indicator of reproductive tract quiescence in pregnant women.” This is a very robust statement that doesn’t necessarily seem supported by the existing data.

Line 199-201: “The findings also strongly suggest that under conditions when vaginal TTV titers are elevated, nutritional requirements in the vaginal milieu are altered such that L. crispatus is no longer capable of preferential proliferation.” There is no evidence to support this claim - it is very speculative, and should be removed or rephrased.

Based on the fact that TTV likely infects lymphoid cells, and that lower immune activation is seen with low-diversity Lactobacillus dominant communities, could it not be equally likely that as the immune system is activated, more lymphocytes arrive at the mucosa and more TTV? So higher TTV is a marker of immune activation, but not at all part of the causal pathway.

Could the authors explain why would it be better to measure TTV than anything else? What does using TTV as a biomarker gain for the scientist or clinician?

“The analysis of vaginal samples for TTV was supported by departmental funds in the Laboratório de Investigaçao Médica em Virologia (LIM-52), Faculdade de Medicina, Universidade de São Paulo. The collection of study samples, microbiome testing and collection of demographic data and pregnancy outcome in this cohort was supported by grants from the Bill and Melinda Gates Foundation, Brazilian Ministry of Health (DECIT) and the Brazilian National Research Council (CNPq – grant 401626/2013-0). Sequence data collection and analyses were performed by the IBEST Genomics Resources Core at the University of Idaho, which is supported in part by NIH COBRE grant P30GM103324.”

Reviewers' comments:

Reviewer's Responses to Questions

**Comments to the Author**

1. Is the manuscript technically sound, and do the data support the conclusions?

Reviewer #1: Partly

Reviewer #2: Yes

Reviewer #3: Yes

2. Has the statistical analysis been performed appropriately and rigorously? 

Reviewer #1: Yes

Reviewer #2: Yes

Reviewer #3: No

3. Have the authors made all data underlying the findings in their manuscript fully available?

Reviewer #1: No

Reviewer #2: Yes

Reviewer #3: No

4. Is the manuscript presented in an intelligible fashion and written in standard English?

Reviewer #1: Yes

Reviewer #2: Yes

Reviewer #3: Yes

5. Review Comments to the Author

Reviewer #1: Main claims and strengths of the paper

The paper is a large study examining the correlation between the vaginal microbiome and Torquetenovirus, matrix metalloproteinases, tissue inhibitors of MMP and lactic acid isomers in 496 pregnant women. The main strength is the size of the cohort, and that it was spread over 3 sites. The results are simple and the conclusions concise. Therefore, the study provides substantial evidence of the inverse link between L. crispatus and TTV titer in the vaginal cavity.

Main weaknesses

In my opinion, the greatest weakness of the paper is a tendency towards over over-interpretation of results. It makes little case for why should be TTV titer should be a superior biomarker to L. crispatus. Since the discussion exclusively discusses the link between L. crispatus and decreased immune activation which is thought to correlate to TTV, it is not clear a qPCR for L. crispatus would not be a much more direct measurement. Similarly, the D-lactic acid and TMP production directly tied to L. crispatus dominance/minority and therefore appear almost redundant. Without measurements of immunological markers, the data only seems to conclude that TTV titter can be used to discriminate a L. crispatus dominated vaginal microbiome from one dominated by G. vaginalis, L. iners or no specific species. I am not sure that the claim in abstract that TTV reflects local conditions can be substantiated, since all measured substance are bacterial metabolites.

Specific comments.

Line 51: Reference needed.

Line 61: I would prefer that you place references right after the relevant text section, rather than grouped together at the end of the section. Also relevant for many other citations in manuscript.

Line 75: Do you have any approval registration numbers etc.? The reader should be able to verify the approval based on the information in the manuscript.

Line 90: I cannot make sense of this sentence. What internal control are you referring to?

Line 106: Even though the methods have been described previously, the reagent kits and sequencing platform should be briefly described for ease of reading.

Line 146: Table 2 was quite unintuitive for me to understand. You compare the L. iners TTV titers to those of other bacteria and find that some are statistically significant from only three of the other dominant bacteria, but this is initially very clear from the figure. However, in the text you write about the positive and negative correlations of bacteria with TTV titer, while Table 2 compares L. iners other bacteria, which is slightly confusing. Possibly a bar plot showing significant pairwise differences would be a better way to present this, while also giving a better idea of variance.

Line 151: The role of MMP’s should be briefly described in the introduction.

Line 165: Were there any correlations between dominant vaginal bacteria and patient characteristics?

Line 177: You write presence/abundance, but in the results section you exclusively use dominant species which is based on abundance. I think presence could be a bit misleading, since the word normally refers to binary presence/absence metrics used in community analysis.

Line 189: I am not sure that you can assume anything about quiescence, as you do not measure immune response or inflammatory markers at all. If you insist on coupling TTV titer with quiescence, it is necessary to include literature references strongly correlating L. iners domination with quiescence.

Line 199: This paragraph is quite vague in its current for and should either be expanded or omitted.

The introduction should have some rationale for why you use L. iners as a reference instead of some other vaginal bacteria. To my knowledge a L. iners dominated vaginal bacterial community is not the only healthy vaginal community state type.

Other comments and questions

Citations should be placed before the full stop consistently.

It should be noted in the discussion that the low sample counts for L. jensenii, gasseri, delbrueckii, vaginalis and Leptopiraceae could be reason that other bacteria are not significantly different. From the median TTV titers, it appears that L. jensenii and gasseri could have a similar inverse correlation as L. crispatus.

I think that the focus of the introduction should be shifted from immune quiescence towards vaginal microbiome, since that is what is measured in the study.

I am unable to find link to the original raw data files from the study, despite this being stated for the submission. I have only had access to the main manuscript file for review, so I apologize if this information is included as supplementary material.

Concluding comment

The study definitely has merit, but the conclusions need to be toned down significantly, and table 2 data should be presented more clearly.

Reviewer #2: Tozetto-Mendoza et al. describe the vaginal TTV loads in relationship with microbiome composition in pregnant women.

The manuscript is of some interest providing new data on a largely unknown topic. Overall, the data are scientifically sound and concisely and clearly described.

Reviewer #3: Abstract

-line 25: missing word: if ‘it’ was predictive

-line 26: Did the authors assess the microbiome or microbiota?

-line 29: definition dominance unclear: does this refer to relative abundance?

-line 36: conclusion unclear – the study describes associations, directionality of these associations and causality cannot be addressed by its design

Introduction

-The referencing style is inconsistent: e.g line 46 individuals.[1,2] vs. line 49 diseases [6,7].

- line 45: Alphatorquetenovirus should be italicised

- it might be helpful to add a paragraph on (i) how TTV are acquired since there’s some literature describing placental acquisition, (ii) where TTV replicates as application appears to occur in T cells which is relevant to the vaginal site and to emphasise (iii) that TTV has previously been suggested as marker of immune status/suppression

-line 50/51: Please add references to support this statement and state for which geography this statement is true. In various parts of the world, pregnancy microbiota composition seems to differ considerably.

-line 56-58: Please add references to support this statement

-line 62-63: Please expand on this section and include other relevant literature, e.g. https://doi.org/10.1086/319983;
https://doi.org/10.1159/000348514; PMID: 12082401 and others

-line 65: Clarify rational of the study as it can only describe associations due to it’s cross-sectional design

Methods

-line 69: it might be helpful to add the gestational age range

-line 91- controls?

-line 103: please change ‘microbiome’ to ‘microbiota’ – this should be corrected throughout the manuscript, as appropriate

-line 106: Please add the public registry and reference number for the project to indicate where the sequencing data is publicly available

-statistics: where analyses adjusted for multiple comparisons? If so, how?

-line 113: There seems to be a word missing in the last sentence. Please add which Graphpad version was used.

Results

-line 116 etc: please add SD or IQR, as appropriate

-line 120: missing comma after available

-Table 1: Is this the BMI at the visit, or pre-pregnancy BMI? I am uncertain how this is relevant to the current study.

-line 132: microbiota

-line 133: clarify the definition of the dominant bacteria: I assume this is based on relative abundance? Were taxa merged on the lowest taxonomic rank that was available, or species level?

-In what proportion of samples was TTV detectable? Have the authors performed analyses based on presence vs. absence of TTV?

-It might be nice to present the results from Tables 2 and 3 visually, rather than as table. P-values could then be added to the graph, and all p-values should be adjusted for multiple comparisons.

-line 141-143: Please report the Spearman rho and not only the p-value.

-line 154: comma missing

-line 165-170: Could these data be added as supplementary data? Where there any differences by site? How did the concentrations of measured MMPs etc relate to the dominant bacterial taxa?

Discussion

-the font type is not consistent throughout the discussion

General comments:

This is an interesting and relevant study that will be of interest to researchers in the field. I have some concerns about the statistical analysis, e.g. the lack of adjusting p-values for multiple comparisons, and the conclusion the authors make based on the available data. It should be emphasised that the authors describe associations and cannot make any conclusions about directionality of events or causality. A thorough proof-reading of the manuscript might be advisable to reduce grammatical errors. Data that is not shown in the manuscript should be included as supplementary material.

6. PLOS authors have the option to publish the peer review history of their article (what does this mean?). If published, this will include your full peer review and any attached files.

Reviewer #1: No

Reviewer #2: No

Reviewer #3: No

---

## [Author Response · Author response to Decision Letter 0]

20 Nov 2021

PONE-D-21-30550

The vaginal Torquetenovirus titer varies with vaginal microbiome composition in pregnant women

Dr. Caroline Mitchell

Academic Editor, PLoS One

Dear Dr. Mitchell,

 We thank you and the reviewers for the favorable review and constructive comments on our manuscript. We have incorporated all of the suggestions into our revision. The specific comments and our responses are as follows:

Editor. 

 I strongly agree with the reviewers' comments that the conclusions are overstated, and several speculative comments are made which are not supported by the data. In a field like this where data are few, I would encourage you to make more factual statements about the findings about correlation between TTV titer and microbial communities, and not speculate about the causal relationship between them - especially based on cross-sectional data. It might be worth a little more discussion of TTV and its link to immune competence, and how you think TTV could be used as a biomarker IF your results are confirmed. 

Answer: The manuscript has been extensively revised so that our conclusions and other statements are more in keeping with the data and are not overly speculative. The Introduction has been expanded to include more information and references on TTV, especially as related to pregnancy. 

Line 64: Goal is to determine “…if the TTV titer in vaginal secretions was predictive of vaginal microbiome composition and pregnancy-related parameters.” 

Answer: This sentence has been deleted and replaced with a more specific aim.

Line 90: “All samples were suitable for DNA amplification by using internal control.” – it is unclear exactly what this means? Was a region amplified to show there was no inhibition? Was the quality and quantity of nucleic acid measured? 

Answer: This is now fully explained in lines 104-106.

Line 180-185: “We propose that a low or undetectable vaginal TTV titer is a biomarker for lower genital tract quiescence during the second trimester of pregnancy, similar to when L. crispatus is dominant at that site.” – How can you separate the quiescence and L. crispatus dominance? It doesnt' seem that you have shown that the two are separately associated with TTV. 

Answer: We agree that this is an example of our over-interpretation of the data. This has been replaced by a more limited and more focused evaluation of the data in the revised manuscript. 

Line 191-193: “Thus, a low vaginal TTV titer appears to parallel L. crispatus vaginal dominance as an indicator of reproductive tract quiescence in pregnant women.” This is a very robust statement that doesn’t necessarily seem supported by the existing data. 

Answer: We agree and have deleted this sentence and replaced it with a more focused interpretation in the revised manuscript. 

Line 199-201: “The findings also strongly suggest that under conditions when vaginal TTV titers are elevated, nutritional requirements in the vaginal milieu are altered such that L. crispatus is no longer capable of preferential proliferation.” There is no evidence to support this claim - it is very speculative, and should be removed or rephrased. 

Answer: This speculative sentence has now been deleted in the revised manuscript. 

Based on the fact that TTV likely infects lymphoid cells, and that lower immune activation is seen with low-diversity Lactobacillus dominant communities, could it not be equally likely that as the immune system is activated, more lymphocytes arrive at the mucosa and more TTV? So higher TTV is a marker of immune activation, but not at all part of the causal pathway. 

Answer: Again, we completely agree with your interpretation and now specifically state this in Discussion. 

Could the authors explain why would it be better to measure TTV than anything else? What does using TTV as a biomarker gain for the scientist or clinician? 

Answer: We do not state that TTV is superior to measurement of the individual bacteria in the vaginal microbiome or to quantitation of vaginal compounds. We now state, based on conclusions from prior TTV-related studies, that the observed associations between TTV and vaginal microbiota and vaginal compounds provides new insights into alterations in the vaginal environment that are present when different bacteria predominate. 

Reviewer #1 

The paper is a large study examining the correlation between the vaginal microbiome and Torquetenovirus, matrix metalloproteinases, tissue inhibitors of MMP and lactic acid isomers in 496 pregnant women. The main strength is the size of the cohort, and that it was spread over 3 sites. The results are simple and the conclusions concise. Therefore, the study provides substantial evidence of the inverse link between L. crispatus and TTV titer in the vaginal cavity. 

Answer: We thank the reviewer for this positive assessment. 

In my opinion, the greatest weakness of the paper is a tendency towards over over-interpretation of results. It makes little case for why should be TTV titer should be a superior biomarker to L. crispatus. Since the discussion exclusively discusses the link between L. crispatus and decreased immune activation which is thought to correlate to TTV, it is not clear a qPCR for L. crispatus would not be a much more direct measurement. Similarly, the D-lactic acid and TMP production directly tied to L. crispatus dominance/minority and therefore appear almost redundant. Without measurements of immunological markers, the data only seems to conclude that TTV titter can be used to discriminate a L. crispatus dominated vaginal microbiome from one dominated by G. vaginalis, L. iners or no specific species. I am not sure that the claim in abstract that TTV reflects local conditions can be substantiated, since all measured substance are bacterial metabolites.

Answer: As stated above we have now greatly modified our interpretations of the actual data. We must disagree with the reviewer as to the origin of MMPs and TIMPs. These compounds are of host origin; only the D- and L-lactic acid are bacterial. Nevertheless, we have modified our conclusion as stated in our last comment to the Editor. 

Line 51: Reference needed. Answer: We have included several references that describe the relative abundance of components of the vaginal microbiota in this revised version. 

Line 61: I would prefer that you place references right after the relevant text section, rather than grouped together at the end of the section. Also relevant for many other citations in manuscript. Answer: We have rearranged the references as recommended.

Line 75: Do you have any approval registration numbers etc.? The reader should be able to verify the approval based on the information in the manuscript. Answer: We now list in Methods specific details of the Institutional Review Board approval. 

Line 90: I cannot make sense of this sentence. What internal control are you referring to? Answer: We apologize for this unclear sentence. We have now greatly clarified our protocol and included a reference in the revised manuscript.

Line 106: Even though the methods have been described previously, the reagent kits and sequencing platform should be briefly described for ease of reading. Answer: This has now been added in the revised manuscript. 

Line 146: Table 2 was quite unintuitive for me to understand. You compare the L. iners TTV titers to those of other bacteria and find that some are statistically significant from only three of the other dominant bacteria, but this is initially very clear from the figure. However, in the text you write about the positive and negative correlations of bacteria with TTV titer, while Table 2 compares L. iners other bacteria, which is slightly confusing. Possibly a bar plot showing significant pairwise differences would be a better way to present this, while also giving a better idea of variance. Answer: Thank you for this comment. We have significantly shortened the Table by combining bacterial species with similar known activities to now make the observed differences between species more easily understandable. We have also expanded the text to include more information in the lines 144 -170. 

Line 151: The role of MMP’s should be briefly described in the introduction. Answer: This has now been added to the Introduction in the revised manuscript.

Line 165: Were there any correlations between dominant vaginal bacteria and patient characteristics? Answer: Since most women at risk for preterm birth were treated with progesterone, as stated in the text, there were no observed differences between bacterial dominance and patient characteristics or outcome. We have now added a statistical analysis that demonstrates no associations between TTV titer and patient characteristics. 

Line 177: You write presence/abundance, but in the results section you exclusively use dominant species which is based on abundance. I think presence could be a bit misleading, since the word normally refers to binary presence/absence metrics used in community analysis. Answer: We agree and have now removed the word “presence” from the manuscript. 

Line 189: I am not sure that you can assume anything about quiescence, as you do not measure immune response or inflammatory markers at all. If you insist on coupling TTV titer with quiescence, it is necessary to include literature references strongly correlating L. iners domination with quiescence. Answer: We agree that use of the word “quiescence “ is speculative and it has now been removed from the text. 

Line 199: This paragraph is quite vague in its current for and should either be expanded or omitted. Answer: We have now omitted this speculation from Discussion.

The introduction should have some rationale for why you use L. iners as a reference instead of some other vaginal bacteria. To my knowledge a L. iners dominated vaginal bacterial community is not the only healthy vaginal community state type. Answer: The Introduction has been extensively modified to clarify what is known about different bacteria in the vaginal microbiota. Actually, L. iners dominance is associated with an unhealthy pregnancy outcome. 

Citations should be placed before the full stop consistently. Answer: This has now been corrected. 

It should be noted in the discussion that the low sample counts for L. jensenii, gasseri, delbrueckii, vaginalis and Leptopiraceae could be reason that other bacteria are not significantly different. From the median TTV titers, it appears that L. jensenii and gasseri could have a similar inverse correlation as L. crispatus. Answer: Thank you for this suggestion that we have now incorporated into study limitations in Discussion. Table 2 has also been modified in response to your suggestion. 

I think that the focus of the introduction should be shifted from immune quiescence towards vaginal microbiome, since that is what is measured in the study. Answer: We agree and have extensively modified the Introduction to now deal mostly with TTV and with the vaginal microbiota. 

I am unable to find link to the original raw data files from the study, despite this being stated for the submission. I have only had access to the main manuscript file for review, so I apologize if this information is included as supplementary material. Answer: The raw data files are available upon request. 

The study definitely has merit, but the conclusions need to be toned down significantly, and table 2 data should be presented more clearly. Answer: Thank you for acknowledging the merit of our study. We have toned down the conclusions considerably and modified Table 2. 

Reviewer #2

Tozetto-Mendoza et al. describe the vaginal TTV loads in relationship with microbiome composition in pregnant women. The manuscript is of some interest providing new data on a largely unknown topic. Overall, the data are scientifically sound and concisely and clearly described. Answer: We thank the reviewer for his kind comments. 

Reviewer #3 

-line 25: missing word: if ‘it’ was predictive. Answer: This has been corrected.

-line 26: Did the authors assess the microbiome or microbiota? Answer: Throughout the text microbiome has now been replaced by microbiota. 

-line 29: definition dominance unclear: does this refer to relative abundance?. Answer: We now specify that dominance refers to relative abundance and this term is now used throughout the text.

-line 36: conclusion unclear – the study describes associations, directionality of these associations and causality cannot be addressed by its design. Answer: We agree and have extensively modified the discussion and the conclusion in this revised version. 

Introduction

-The referencing style is inconsistent: e.g line 46 individuals.[1,2] vs. line 49 diseases [6,7]. Answer: The reference style is now uniform throughout the manuscript. 

- line 45: Alphatorquetenovirus should be italicized. Answer: This has been done.

- it might be helpful to add a paragraph on (i) how TTV are acquired since there’s some literature describing placental acquisition, (ii) where TTV replicates as application appears to occur in T cells which is relevant to the vaginal site and to emphasise (iii) that TTV has previously been suggested as marker of immune status/suppression. Answer: The Introduction has been modified to include more information about TTV as requested. 

-line 50/51: Please add references to support this statement and state for which geography this statement is true. In various parts of the world, pregnancy microbiota composition seems to differ considerably. Answer: This statement has been modified and references added. 

-line 56-58: Please add references to support this statement. Answer: References have been rearranged to more appropriate places in the manuscript. 

-line 62-63: Please expand on this section and include other relevant literature, e.g. https://doi.org/10.1086/319983;
https://doi.org/10.1159/000348514; PMID: 12082401 and others. Answer: The suggested reference plus others on TTV in pregnancy have been added. 

-line 65: Clarify rational of the study as it can only describe associations due to it’s cross-sectional design. Answer: We agree and have modified and now more clearly describe the aims of the study. 

Methods

-line 69: it might be helpful to add the gestational age range. Answer: This is provided in Table 1. 

-line 91- controls? Answer: This has now been extensively clarified and a new reference has been added. 

-line 103: please change ‘microbiome’ to ‘microbiota’ – this should be corrected throughout the manuscript, as appropriate. Answer: This has been done as suggested.

-line 106: Please add the public registry and reference number for the project to indicate where the sequencing data is publicly available. Answer: References are provided in the text for the sequences used in our study. None of the sequencing data is original but has been used numerous times by other investigators. 

-statistics: where analyses adjusted for multiple comparisons? If so, how? Answer: The analyses were not adjusted for multiple comparisons since the number of comparisons is small and the p values in all comparisons were highly significant. 

-line 113: There seems to be a word missing in the last sentence. Please add which Graphpad version was used. Answer: This has now been added.

Results

-line 116 etc: please add SD or IQR, as appropriate. Answer: This information is provided in the Table and it seemed to be redundant to also include it in the text. 

-line 120: missing comma after available. Answer: This has been corrected.

-Table 1: Is this the BMI at the visit, or pre-pregnancy BMI? I am uncertain how this is relevant to the current study. Answer: BMI refers to BMI at the time of the second trimester visit. Its inclusion is just to provide a more complete characterization of the patients. 

-line 132: microbiota. Answer: This has been changed.

-line 133: clarify the definition of the dominant bacteria: I assume this is based on relative abundance? Were taxa merged on the lowest taxonomic rank that was available, or species level?

-In what proportion of samples was TTV detectable? Have the authors performed analyses based on presence vs. absence of TTV? Answer: We have now clarified that dominant bacteria refers to the situation where the relative abundance of a specific bacterium was >50% of the total number of bacteria identified. We have modified Table 2 to now also present data on the species level. We have now included a sentence in Results describing the percent of cases in which the TTV level was below our lower limit of detection. We have not performed a separate analysis based on the detection or non-detection of TTV. We did not think this was would provide additional valid information since it would eliminate a different percentage of values for each bacterium. 

-It might be nice to present the results from Tables 2 and 3 visually, rather than as table. P-values could then be added to the graph, and all p-values should be adjusted for multiple comparisons. Answer: As mentioned in our comment to Reviewer 1 Table 2 has been extensively revised to make the data more easily understandable. 

-line 141-143: Please report the Spearman rho and not only the p-value. Answer: This has now been added as requested. 

-line 154: comma missing. Answer: This has been added.

-line 165-170: Could these data be added as supplementary data? Where there any differences by site? How did the concentrations of measured MMPs etc relate to the dominant bacterial taxa?

 Answer: We have now added a statistical analysis of all these data. 

Discussion

-the font type is not consistent throughout the discussion. 

Answer: This has been corrected.

General comments:

This is an interesting and relevant study that will be of interest to researchers in the field. I have some concerns about the statistical analysis, e.g. the lack of adjusting p-values for multiple comparisons, and the conclusion the authors make based on the available data. It should be emphasised that the authors describe associations and cannot make any conclusions about directionality of events or causality. A thorough proof-reading of the manuscript might be advisable to reduce grammatical errors. Data that is not shown in the manuscript should be included as supplementary material. 

Answer: We have addressed all the issues mentioned by the reviewer and the text has been significantly modified to reduce grammatical errors and reduce the amount of speculation. 

We are available to send you any further information you may require.

Yours faithfully,

Tania Regina Tozetto-Mendoza and co-authors

Laboratory of Virology,

Institute of Tropical Medicine

School of Medicine, University of São Paulo, Brazil

---

## [Decision Letter · Decision Letter 1]

15 Dec 2021

PONE-D-21-30550R1The vaginal Torquetenovirus titer varies with vaginal microbiota composition in pregnant womenPLOS ONE

Dear Dr. Tozetto-Mendoza,

Thank you for submitting your manuscript to PLOS ONE. After careful consideration, we feel that it has merit but does not fully meet PLOS ONE’s publication criteria as it currently stands. Therefore, we invite you to submit a revised version of the manuscript that addresses the points raised during the review process.

We look forward to receiving your revised manuscript.

Kind regards,

Caroline Mitchell

Academic Editor

PLOS ONE

Additional Editor Comments (if provided):

I agree with the reviewer's comments about the difficulty in assessing association with pregnancy outcomes when some people were treated with progesterone, but we have no information about who those people were. If discussing pregnancy outcomes, please present metadata and TTV data that allow more robust interpretation. Table 1 could be presented with 3 columns: term, sPTB, iPTB and could include those data.

Additionally, both reviewers have highlighted areas for clarification which should be considered.

Reviewers' comments:

Reviewer's Responses to Questions

**Comments to the Author**

1. If the authors have adequately addressed your comments raised in a previous round of review and you feel that this manuscript is now acceptable for publication, you may indicate that here to bypass the “Comments to the Author” section, enter your conflict of interest statement in the “Confidential to Editor” section, and submit your "Accept" recommendation.

Reviewer #1: All comments have been addressed

Reviewer #3: (No Response)

2. Is the manuscript technically sound, and do the data support the conclusions?

Reviewer #1: Yes

Reviewer #3: No

3. Has the statistical analysis been performed appropriately and rigorously? 

Reviewer #1: Yes

Reviewer #3: No

4. Have the authors made all data underlying the findings in their manuscript fully available?

Reviewer #1: No

Reviewer #3: No

5. Is the manuscript presented in an intelligible fashion and written in standard English?

Reviewer #1: Yes

Reviewer #3: No

6. Review Comments to the Author

Reviewer #1: I think that the authors have done a good job, and that the revised manuscript has been greatly improved by the streamlining and more substantiated conclusions. There are still 2 points that I think should be adressed before acceptance:

1) All supplementary material should be uploaded as suuplementary files along with the manuscript or added to an online repository such as Zotero

2) Multiple testing should still be performed where appropriate, even though it will not change the conclusions, as its absence is disctracting form the conclusion.

I also have the following minor corrections:

Line 66: Missing word? ”…, but the aetiology still remains poorly understood” for example ?

Line 78: Do you mean abundance above a certain level, or that abundance correlates with decreased risk of preterm birth?

Line 82: “Whether” rather than “if”?

Line 107: It is still unclear what is meant what is meant by “Suitability of the DNA for amplification”. I assume that it refers to lack of PCR inhibitors?

Line 124: Please refer to original article using the method, since current reference also states “as previously described”. As far as I can see this is the correct reference. https://doi.org/10.1038/s41598-017-09857-z . Sequencing platform should still be stated in the abbreviated description.

Reviewer #3: The data availability statement states that all data are included in the manuscript; however, in the letter to the editor the authors state ‘The raw data files are available upon request.“ Please clarify.

Abstract

Line 32: L. iners was dominant: L. iners should be italicised, ‘were’ instead ‘was’

“There was no association between TTV titre and pregnancy outcome.” However, you state in your letter to the editor that ‘most women at risk for preterm birth were treated with progesterone’ – If this is true, the authors cannot conduct this analysis and the conclusion is misleading.

“We conclude that quantitation of the TTV titer in vaginal secretions during the mid-trimester of pregnancy reflects the relative abundance of specific bacterial species and vaginal compounds at that site.” Please clarify that TTV titre was associated with relative abundance. Further, what is your conclusion? What does it meant that TTV titre is inversely or positively associated with specific bacteria?

Introduction

Line 63: or should not be italicised

Line 83: I do not understand how the aims of the study was ‘to identify associations with pregnancy outcome“ if the authors included women who were treated with progesterone to avert PTB. Where these women still included in the analysis?

Methods

What were the enrolment criteria? If associations with pregnancy outcome were evaluated, these should be outlined in more detail.

Line 107-108: Please explain your control more. Beta-globin is a human gene while TTV is a virus. I am uncertain how this assessment would confirm stability or viral nucleic acids?

Line. 124-127: Please expand this section. Which specific primers were used? Was data analysed using DADA2? Which database was used for taxonomic assignment? Which other downstream analysis was conducted? Were extraction controls included? How did the authors control for potential contamination?

Results:

Were p-values adjusted for multiple comparisons? The Spearman correlations will always be influenced by the large sample size – Just because a p-value is significant does not meant that a Spearman correlation of 0.1 biologically meaningful.

Table 2: Bacterial species should be italicised

Line 193-195: I do not think that this comparison is valid, given that some women in the cohort were treated when at high risk for PTB.

Line 231: “In addition, the range of TTV titers that we identified in the vagina parallels the levels reported to be present in peripheral blood (2-8 log10 copies/ml) [28], validating our TTV protocol.” How does this validate your protocol? Would you expect that systemic and vaginal viral titres are identical?

In summary, it is unclear to me what the manuscript in its current form adds to the literature. The study is correlative in nature, and the observed correlations are not particularly strong. Further, the assessment with birth outcomes seems to be inaccurate, given that some women received treatment to prevent PTB.

7. PLOS authors have the option to publish the peer review history of their article (what does this mean?). If published, this will include your full peer review and any attached files.

Reviewer #1: No

Reviewer #3: No

---

## [Author Response · Author response to Decision Letter 1]

31 Dec 2021

Academic Editor

Dear Dr. Mitchell,

 We thank you and the reviewers for helpful comments on our revised submission, ONE-D-21-30550R1 “The vaginal Torquetenovirus titer varies with vaginal microbiota composition in pregnant women”. The individual comments and are responses are detailed below. 

1. I agree with the reviewer's comments about the difficulty in assessing association with pregnancy outcomes when some people were treated with progesterone, but we have no information about who those people were. If discussing pregnancy outcomes, please present metadata and TTV data that allow more robust interpretation. Table 1 could be presented with 3 columns: term, sPTB, iPTB and could include those data.

Response: We completely agree with your comment and those of the reviewer that we are not in a position to accurately evaluate the association between TTV titer and pregnancy outcome. Providing progesterone treatment to women deemed to be at high risk for a preterm birth introduced another variable (uncontrolled) that made conclusions about a relationship between vaginal TTV titer in the second trimester and risk of premature delivery in our population difficult to interpret. We acknowledge that the relation between TTV and premature delivery must of necessity be studied separately in a prospective investigation in which all subjects receive the same treatment. Therefore, in our revision we have now removed as an aim of our study to determine the association between TTV and pregnancy outcome. However, we have now added data in Results showing that there is no difference in the mid-trimester TTV titer between women who were subsequently treated with progesterone (and therefore deemed to be at elevated risk for preterm labor) and untreated women (not at high risk). This is a more accurate statement of what we actually determined. We would like to stress that the vaginal samples from all subjects were obtained prior to progesterone administration, ensuring that this procedure did not influence the vaginal TTV titer, microbiome composition or the concentration of compounds in vaginal fluid. The remaining aims of the study and our conclusion in defining associations between TTV titer, the dominance of individual bacteria in the vaginal microbiota and concentrations of compounds in vaginal fluid remain valid and we believe are unique and worthy of dissemination.

Reviewer’s Comments to the Author

Reviewer #1: I think that the authors have done a good job, and that the revised manuscript has been greatly improved by the streamlining and more substantiated conclusions. There are still 2 points that I think should be addressed before acceptance:

1) All supplementary material should be uploaded as supplementary files along with the manuscript or added to an online repository such as Zotero. 

Response:Three Tables and a Figure have now been uploaded with the manuscript under the title Supplementary Files (Supporting information).

2) Multiple testing should still be performed where appropriate, even though it will not change the conclusions, as its absence is distracting form the conclusion. Response: In the revision we have changed the statistical evaluations to now include a modified analysis that accounts for the influence of multiple testing on the reported associations, using the Bonferroni correction. The p value for a positive association has now been set at 0.005.

I also have the following minor corrections:

Line 66: Missing word? ”…, but the aetiology still remains poorly understood” for example ? Response: This has been corrected.

Line 78: Do you mean abundance above a certain level, or that abundance correlates with decreased risk of preterm birth? 

Response: As mentioned above we have now removed from the text statements related to pregnancy outcome. 

Line 82: “Whether” rather than “if”?

 Response:This has been changed as requested. 

Line 107: It is still unclear what is meant what is meant by “Suitability of the DNA for amplification”. I assume that it refers to lack of PCR inhibitors? 

Response:In the revision we clarify the statement further. What we meant to convey and now explicitly state was that our ability to amplify the control gene indicates that the DNA was not degraded and there were no PCR inhibitors in the preparations. 

Line 124: Please refer to original article using the method, since current reference also states “as previously described”. As far as I can see this is the correct reference. https://doi.org/10.1038/s41598-017-09857-z . Sequencing platform should still be stated in the abbreviated description.

 Response: We thank the reviewer for these comments. We apologize for using an incorrect reference, This has now been corrected as suggested and we now also have increased our description of the protocol, including the sequencing platform utilized. 

Reviewer #3: The data availability statement states that all data are included in the manuscript; however, in the letter to the editor the authors state ‘The raw data files are available upon request.“ Please clarify. 

Response: We have now included all remaining experimental data as Supplementary files (supporting information).

Abstract

Line 32: L. iners was dominant: L. iners should be italicised, ‘were’ instead ‘was’ 

Response:These have been corrected.

“There was no association between TTV titre and pregnancy outcome.” However, you state in your letter to the editor that ‘most women at risk for preterm birth were treated with progesterone’ – If this is true, the authors cannot conduct this analysis and the conclusion is misleading. 

Response: We completely agree and have now removed from the manuscript pregnancy outcome as an aim of our study. Please see our comment to the Editor above for a full explanation. Again, we wish to stress that the removal of pregnancy outcome data and their analysis does not reduce the value of our findings that the dominant bacteria in the vaginal microbiota and the concentration of compounds in vaginal fluid varies with the vaginal TTV titer. This novel finding furthers our understanding of differential bacteria domination in the vaginal microbiome in individual women and, hopefully, will eventually lead to development of protocols to deliberately alter the vaginal milieu to maximize women’s health.

“We conclude that quantitation of the TTV titer in vaginal secretions during the mid-trimester of pregnancy reflects the relative abundance of specific bacterial species and vaginal compounds at that site.” Please clarify that TTV titre was associated with relative abundance. Further, what is your conclusion? What does it meant that TTV titre is inversely or positively associated with specific bacteria? 

Response: We thank the reviewer for these comments. Our conclusion is now been clarified to it more accurate and more easily understandable. “We conclude that variations in the TTV titer with the relative abundance of different bacterial species and concentration of vaginal compounds is consistent with the local immune status being an influence on vaginal fluid composition.” 

Introduction

Line 63: or should not be italicized. Response: This has been corrected.

Line 83: I do not understand how the aims of the study was ‘to identify associations with pregnancy outcome“ if the authors included women who were treated with progesterone to avert PTB. Where these women still included in the analysis? Response: We completely agree. This statement has now been deleted. A complete explanation for doing this is provided above. 

Methods

What were the enrolment criteria? If associations with pregnancy outcome were evaluated, these should be outlined in more detail. Response: Again, we have deleted from the text an analysis and discussion of pregnancy outcome. The enrolment criteria are now fully described in Methods. 

Line 107-108: Please explain your control more. Beta-globin is a human gene while TTV is a virus. I am uncertain how this assessment would confirm stability of viral nucleic acids? Response:We have now expanded this section as follows. Suitability of the DNA for viral DNA amplification was evaluated by first testing the samples for amplification of the gene coding for beta-globin. This was successful in every case and indicated that the DNA was intact and that our preparations did not contain PCR inhibitors. In addition, specifically concerning the TTV DNA, we obtained a suitable amplification profile by real time PCR in every TTV positive samples and controls to validate that the viral DNA was quantitatively stable. This data is now included as a supplementary fig.1 (supporting information). 

Line 124-127: Please expand this section. Which specific primers were used? Was data analysed using DADA2? Which database was used for taxonomic assignment? Which other downstream analysis was conducted? Were extraction controls included? How did the authors control for potential contamination? ?? Response: We thank the reviewer for these comments and for the opportunity to clarify this section. As mentioned in our response to the comment of Reviewer #1 we incorrectly referenced one of our subsequent studies and not our original reference. This has been corrected. Description of this analysis have now been greatly expanded. However, we would like to emphasize that details of our microbiome analyses have been already published and all relevant controls can be found in our included references. 

Results:

Were p-values adjusted for multiple comparisons? The Spearman correlations will always be influenced by the large sample size – Just because a p-value is significant does not meant that a Spearman correlation of 0.1 biologically meaningful. Response: In the revision we now state in Methods that the p value for a positive association has been adjusted for multiple comparisons by employing the Bonferroni correction. The p value for a positive association is now set at <0.005. 

Table 2: Bacterial species should be italicized. Response: This has been corrected.

Line 193-195: I do not think that this comparison is valid, given that some women in the cohort were treated when at high risk for PTB. Response: We again stress that when progesterone was given it was always after the collection of vaginal samples for analysis. Therefore, this treatment cannot influence our results. 

Line 231: “In addition, the range of TTV titers that we identified in the vagina parallels the levels reported to be present in peripheral blood (2-8 log10 copies/ml), validating our TTV protocol.” How does this validate your protocol? Would you expect that systemic and vaginal viral titres are identical? Response: We agree that this statement was overly ambitious and not based on actual data. The sentence has now been deleted and replaced by the sentence, “It is still of interest to point out that the range of TTV titers we identified in the vagina parallel the levels reported to be present in peripheral blood (2-8 log10 copies/ml).

In summary, it is unclear to me what the manuscript in its current form adds to the literature. The study is correlative in nature, and the observed correlations are not particularly strong. Further, the assessment with birth outcomes seems to be inaccurate, given that some women received treatment to prevent PTB. Response: We completely agree, as discussed exhaustively above, that our assessment with birth outcomes may be inaccurate and, thus, this part of the manuscript has been deleted. However, we disagree that the manuscript does not substantially add anything new to the literature. Perhaps we did not stress strongly enough or clarify sufficiently the value of our observations. Specifically, the association between variations in the TTV titer and the relative abundance of specific bacterial species and vaginal compounds indicates that local changes in immune status likely influence vaginal fluid composition. This has now been made more explicit in the conclusion section of Discussion. 

We are available to send you any further information you may require.

Yours faithfully,

Tania Regina Tozetto-Mendoza and co-authors

---

## [Editor Report · Decision Letter 2]

4 Jan 2022

The vaginal Torquetenovirus titer varies with vaginal microbiota composition in pregnant women

PONE-D-21-30550R2

Dear Dr. Tozetto-Mendoza,

We’re pleased to inform you that your manuscript has been judged scientifically suitable for publication and will be formally accepted for publication once it meets all outstanding technical requirements.

Kind regards,

Caroline Mitchell

Academic Editor

PLOS ONE
---

## [Editor Report · Acceptance letter]

10 Jan 2022

PONE-D-21-30550R2 

The vaginal Torquetenovirus titer varies with vaginal microbiota composition in pregnant women 

Dear Dr. Tozetto-Mendoza:

I'm pleased to inform you that your manuscript has been deemed suitable for publication in PLOS ONE. Congratulations! Your manuscript is now with our production department. 

Kind regards, 

on behalf of

Dr. Caroline Mitchell 

Academic Editor

PLOS ONE